# Deep Learning Approaches for Robust Time of Arrival Estimation in Acoustic Emission Monitoring

**DOI:** 10.3390/s22031091

**Published:** 2022-01-31

**Authors:** Federica Zonzini, Denis Bogomolov, Tanush Dhamija, Nicola Testoni, Luca De Marchi, Alessandro Marzani

**Affiliations:** 1Advanced Research Center on Electronic Systems “Ercole De Castro” (ARCES), University of Bologna, 40136 Bologna, Italy; denis.bogomolov2@unibo.it (D.B.); nicola.testoni@unibo.it (N.T.); 2Department of Electrical, Electronic and Information Engineering (DEI), University of Bologna, 40136 Bologna, Italy; tanush.dhamija@studio.unibo.it (T.D.); l.demarchi@unibo.it (L.D.M.); 3Department of Civil, Chemical, Environmental and Materials Engineering (DICAM), University of Bologna, 40136 Bologna, Italy; alessandro.marzani@unibo.it

**Keywords:** acoustic emissions, capsule neural network, convolutional neural network, source localization, time of arrival

## Abstract

In this work, different types of artificial neural networks are investigated for the estimation of the time of arrival (ToA) in acoustic emission (AE) signals. In particular, convolutional neural network (CNN) models and a novel capsule neural network are proposed in place of standard statistical strategies which cannot handle, with enough robustness, very noisy scenarios and, thus, cannot be sufficiently reliable when the signal statistics are perturbed by local drifts or outliers. This concept was validated with two experiments: the pure ToA identification capability was firstly assessed on synthetic signals for which a ground truth is available, showing a 10× gain in accuracy when compared to the classical Akaike information criterion (AIC). Then, the same models were tested via experimental data acquired in the framework of a localization problem to identify targets with known coordinates on a square aluminum plate, demonstrating an overreaching precision under significant noise levels.

## 1. Introduction

Among the non-destructive testing methods for structural health monitoring (SHM), the one based on acoustic emission (AE) is particularly effective for the assessment of civil infrastructure and industrial plants, allowing the detection of active damages in structures such as buildings and bridges, pipelines, storage tanks, etc. AE testing is built on the analysis of the acoustic activity of the target structure [1], primarily due to the growth of cracking phenomena. One of the main advantages of AE relies on the possibility to localize such sources by passively capturing the acoustic response of the structure. It is, therefore, from the extraction of a batch of representative acoustic features, typically defined on a time basis, and their evolution over time, that potential dangerous defects can be detected at an early stage of degradation, and preventive alarms can be issued [2]. One of the important parameters to be extracted from a detected acoustic signal consists of the time of arrival (ToA), also known as onset time, namely the time taken by the induced wave to travel from its origin to the AE transducer. ToA, when assessed from multiple sensing positions, provides a means to localize the AE signal source. The literature about these localization methods is quite vast and comprehends, among others, approaches based on geometrical or angular relationships, such as the ones built on the angle of arrival or the difference time of arrival (DToA) [3]. Alternative strategies are based on the estimation of the signal energy [4], but this parameter is typically very sensitive to environmental and operational factors (e.g., imperfect coupling between the AE transducers and the structure), which hamper their exploitation for long-term and continuous integrity evaluation.

AE testing is typically implemented with long inspection intervals by purposely increasing the mechanical load applied to the structure [5] (https://eur-lex.europa.eu/legal-content/IT/TXT/?uri=celex:32014L0068 (accessed on Tuesday, 28 December 2021), https://ec.europa.eu/docsroom/documents/41641 (accessed on Tuesday, 28 December 2021)). However, more recently, a new approach has emerged, in which AE monitoring is performed in real time, via the permanent placement of AE equipment on the monitored structure and detecting the emissions generated during the normal load cycles. Despite the advantages in terms of responsiveness and preventive detection, the adoption of the latter approach is challenged by the difficulty in accurately identifying weak AE events in noisy environments, particularly in the early stages of defect growth [6]. Indeed, permanently installed AE systems must counteract the corruption of noise generated by operational processes, for example, vibrations of rotating machinery (e.g., a pump unit or an engine), responsible for disturbances in the surrounding environment [7] and of unpredictable noise sources, such as electromagnetic interference and ambient noise in the vicinity of the monitoring system [8,9].

The less favorable signal-to-noise ratios (SNRs), which characterize real-time monitoring acquisitions, may prevent the accurate tracking of AE features, and in particular, of the ToA. Statistical algorithms for onset time determination have proven their effectiveness at high SNRs, but the noise factors mentioned above might significantly affect the reproducibility and accuracy of ToA estimation results. Consequently, the implementation of reliable methods for signal detection, robust against operational noisy environments, is still an open research field.

The objective of the novel methods proposed in this article is (i) to build accurate and alternative models capable of handling heavily corrupted AE signals and (ii) to test them in operative SHM frameworks, such as acoustic source localization purposes. In particular, we considered the case of AE propagating in waveguides, so that the onset detection difficulty is further exacerbated by dispersion and multi-modality, as discussed in the following subsection.

### 1.1. Acoustic Emissions in Waveguides

When an acoustic emission event occurs as a consequence of crack, corrosion or delamination processes in a waveguide, ultrasonic guided waves (GWs) are generated and can travel long distances [10,11].

Lamb waves are the particular form of GWs which propagate in plates. Albeit showing long-range propagation and sensitivity with respect to distinct classes of damage, Lamb waves can exhibit complex behavior during propagation due to their multi-modal and dispersive nature. *Multi-modal* means that multiple Lamb waves, or guided modes, co-exist in the same frequency interval (see Figure 1). These modes are denoted as symmetric (S) (red curves) or anti-symmetric (A) (red curves) depending on the nature of the wavefield with respect to the mid plane of the plate. *Dispersive* means that Lamb waves are characterized by a frequency-dependent wave speed.

The number and type of the generated modes vary with the frequency and shape of the actuation. In general, it is possible to design the actuation in time and space to limit the effect of multi-modality and dispersion in order to reduce the complexity of the generated/received signals. In more detail, Figure 1 shows the dispersion curves [12] in terms of phase $c_{p}$ (left panel) and group $c_{g}$ (right panel) velocity of the Lamb waves existing in an aluminum plate of infinite extension and 1 mm thickness.

In practical cases, where it is necessary to deal with a finite plate, other complexities arise, such as the physical interaction of GWs with the mechanical boundaries, which is responsible for reflections and reverberations, a phenomenon also known as multi-path interference, the mutual interference in the case of multiple active transducers, and also the effects of environmental changes, such as temperature fluctuations, that can alter the wave propagation behavior.

### 1.2. ToA Estimation: From Statistical Methods to Machine Learning

By computing the similarity between two signals, cross correlation (X-Corr) can be used as a powerful tool for the estimation of the time shift of two time series [13]. In practical AE scenarios, where the monitoring network consists of a passive mesh of transducers and the true excitation source is unknown, X-Corr offers a means for DToA quantification among pairs of receivers, rather than a measure of the actual ToA. It has proved its effectiveness for the characterization of AE signals in multiple environments [14]. Combining X-Corr and frequency warping is an effective means to tackle dispersion and multi-modality in guided propagation [15]. However, X-Corr is highly susceptible to minor perturbations in the statistical properties of the input signals (e.g., residual noise sources superposed on the actual information content).

To cope with these issues, the Akaike information criterion (AIC) approaches the task of ToA identification as a pure statistical problem based on the second order statistics of the measured acoustic data. In essence, AIC leverages the concept of signal entropy to detect abrupt changes in the statistical distribution of the observed signal *y* [16,17]. For a discrete signal with *N* samples, this is achieved by computing, for each sample *k*, the quantity
(1)$$\mathrm{AIC}\left[k\right]=k\log\sigma_{y[1:k]}^{2}+\left(N-k-1\right)\log\sigma_{y[k+1:N]}^{2}$$
which is a logarithmic measure of the cumulative variance ($\sigma_{y}^{2}$) of the preceding (y[1:k]) and successive (y[k+1:N]) signal window with respect to the current sample index *k*. In other words, AIC splits the full waveform into a *k* dimensional and an N−k dimensional smaller vector, respectively spanned by the first *N* and the last N−k samples, and describes the level of similarity between them. The rationale is that, in correspondence of a sharp change in the signal profile, such as the one associated with the arrival of the incoming wave-front generated by the acoustic source, the divergence between the two variances increases to a large extent, generating a minimum in the overall AIC function. The time instant aligned with this minimum is the sought ToA [17].

Notably, this method can provide reliable outcomes when the processed signal presents two clearly distinct regions, e.g., a high-entropy portion where uncorrelated noise dominates, and a low-entropy segment where the acoustic signal is present [18]. Nevertheless, as discussed in [17], this might not be the case for AE monitoring scenarios, in which the mere attenuation due to signal propagation, which is responsible for low-amplitude received waveforms, is further hindered by additive operative noise, demanding for more advanced data processing solutions.

Led by the constantly increasing success of artificial intelligence (AI) in learning complex patterns hidden within signals, interesting AI solutions to tackle ToA estimation have been proposed, with particular emphasis on the seismology field. For example, a template-based artificial neural network (NN) for earthquake phase detection was proposed in [19], while [20] proposed an unsupervised fuzzy clustering logic for ToA recognition in micro-seismic waves. Another example worthy of attention was examined in the work by Zachary E. Ross [21], where the ToA of pressure waves in seismograms was considered a pattern recognition problem on top of which machine learning (ML)/deep learning (DL) models were trained. Considered among the most powerful architectures for deep and ultra-deep learning, PhaseNet and U-Net [22] were also investigated for seismic arrival time picking, reporting outstanding results. A comprehensive list of the most recent trends in this direction can be found in [23].

A close analogy exists between the seismic application domain and AEs detection. By virtue of this similarity, we drew inspiration from the solutions tested in seismology to develop the new DL solutions for AE signal processing, which are described and validated in the following sections of this manuscript.

## 2. Deep Learning Models for ToA Estimation

Two different neural network models were implemented for the purpose of ToA estimation, which are presented hereinafter.

### 2.1. Convolutional Neural Network

The convolutional neural network (CNN) [24] is a class of artificial neural networks that can extract relevant information from raw data and retain it in the form of weights and biases of the corresponding layers: the learned parameters are then used to make classifications and/or predictions as soon as new signal instances are available. A general CNN architecture consists of two main stages, i.e., *feature extraction* and *classification*, which are completely described by the following architectural blocks:*Convolutional layers*: it is in charge of feature extraction from input data, which are passed in a tensor form of dimensions $N_{in}$. A convolutional layer runs dot products between the input data and a specific set of weights (or mask), which are stored as taps of a corresponding filter, also known as *kernel*, of dimension $N_{ks}$. This filter is recursively applied to subsequent portions, or patches, of the input data by means of a sliding filter mask, which is shifted by a constant quantity called stride ($N_{stride}$).To increase the learning capability of the network, more than one filter is employed in a single convolutional layer: if $N_{filter}$ is the number of total different kernels per layer, $N_{filter}$ different maps of the the same input data are provided in the output via a proper activation function. A convolutional layer is, thus, completely determined by the tuple of values: $\left(N_{in},N_{filter},N_{ks},N_{stride}\right)$.Multiple convolutional layers are usually stacked one after the other, whose typology is dictated, in turn, by the dimensions of the manipulated data. In the case of ToA estimation, where the problem is intrinsically mono-dimensional and thought to be performed on a sensor-wise basis, 1D convolutional (Conv1D) layers are necessary.*Pooling layers*: this layer provides a distilled version of each feature map to shrink the computational complexity and the spatial size of the convolved features. Indeed, since the number of points in each feature map returned at the end of a single convolutional block might be extremely large, and also since many of them only capture minor details, they can be neglected. Different pooling strategies have been proposed: max pooling (MaxPool), which only preserves the maximum value in a specific patch of the feature map; and average pooling (AvgPool), which extracts a single scalar as the average of the points falling in the same feature patch.*Dense layers*: once manifold representations are obtained, the sought pattern hidden within them is learned via dense fully connected (FC) layers, i.e., feed-forward layers with neurons that have full connections to all activations delivered by the previous layer. Firstly, a flattening operation is performed to unroll the feature maps provided by the last pooling layer in a uni-dimensional vector of appropriate dimension; then, these values are used as input of a standard artificial neural network, which acts either as a classifier or a regressor, depending on the desired task.

#### 2.1.1. Large CNN Model

The first CNN architecture considered in this work is schematically represented in Figure 2 and is devised for ToA retrieval from 5000-long time series. As can be seen, five ($5000/2^{2l}$,$N_{filter}$,10,2) Conv1D layers (l∈{0,⋯,4} being the layer index), with $N_{filter}\in \left\{50,100,150,200,250\right\}$ and ReLU activation function, are stacked in cascade and followed by a MaxPool layer with a compression factor equal to 2. A global average pooling layer (Global AvgPool) is also included at the end of the convolutional block to force the regressor behavior of the network: Global AvgPool yields one single feature map out of the 250 different representations at the end of the last AvgPool layer. This single map is then passed to a first FC layer having 1024 neurons activated by ReLU; ToA can finally be retrieved from the output layer consisting of a 1×1 FC layer with one neuron and linear activation. It is worth saying that the so-far designed CNN model is characterized by 1,259,299 parameters, requiring a minimum memory space of at least 1.5 MB even in quantized form: the Adam optimizer [25] with learning rate of 0.001 and loss weight equal to 1 was used for training such parameters, and the model was trained for 15 epochs.

#### 2.1.2. Small CNN Model

It must be emphasized the fact that, in order to be applicable in permanent AE equipment installations with custom sensors and electronics compatible with long-term and real-time functionalities, the devised AI solutions must run on low-cost and resource-constrained devices. However, fitting the previously described CNN model to the limited capabilities of edge sensors is not practicable due to the excessive amount of memory (and, in turn, computational power) it requires. To this end, it is paramount to emphasize that the typical static and volatile memory of medium-to-high embedded devices hardly exceeds 1 MB (usually amounting to hundreds of kB) to harvest space and dynamic power consumption (https://www.st.com/en/microcontrollers-microprocessors/stm32-ultra-low-power-mcus.html (accessed on Wednesday, 29 December 2021), https://www.espressif.com/en/products/modules/esp32 (accessed on accessed on Wednesday, 29 December 2021)).

For this reason, a distilled version of the preceding model is derived as displayed in Figure 3, in which the five convolutional layers are substituted with four smaller size Cov1D+MaxPool layers with 16, 32, 64 and 64 filters while leaving unaltered all the remaining parameters. The dimensions of the GlobalAvg layer were changed accordingly. In this lighter version, only 134,481 parameters need to be learned, for a total memory occupancy of nearly 150 kB after conversion to embedded programming format, which leads not only to a complexity reduction of more than 10×, but makes the model absolutely compatible with the above-mentioned memory constraints of edge devices. Hereinafter, to differentiate the two models, this smaller one is called “small CNN”.

Notably, the model reduction of both the number of convolutional layers and filters per layer is preferred over other pruning strategy, given its proven advantages in terms of algorithmic complexity and memory footprint, as well as for its robustness against model over-parametrization [26] and better generalization to out-of-distribution data, as demonstrated in Section 3.3.

### 2.2. Capsule Neural Network

Despite their outstanding performance in multiple fields, CNNs might be ineffective under the following circumstances [27]: (i) the observed data pattern presents shifts/rotations, since CNNs are phase and shift invariant; (ii) the spatial relationship between the feature maps is an important indicator of the data distribution, since CNNs do not exploit spatial dependencies; (iii) the loss of information introduced by the pooling layers is unacceptable, especially for very deep model where pooling is mandatory.

Very recently released in the field of AI [28], CapsNet represents a powerful competitor to convolutional architectures for classification tasks. Three main reasons can be mentioned. Firstly, CapsNet transforms basic feature maps in correlated feature maps via the novel concept of capsule unit: this correlation-based approach implies the preservation of spatial dependencies between data. Secondly, albeit disregarding the pooling layer, it is capable of correct prediction, even when trained on fewer data. Thirdly, its vector-based output allows for robust classification performance by making use of simpler network architectures, favorable for implementation on edge sensors. This latter aspect is owed to the fact that CapsNet actually offers a first means for knowledge distillation, which is performed directly at an architectural level via novel machine learning operators, rather than being executed at a coding/firmware level, where most of the effort is usually spent.

Successful application of CapsNet for micro-seismic phase picking was accounted in [27], showing great performance for earthquake signal characterization. Inspired by this first attempt, an AE-oriented variant of CapsNet is proposed in this manuscript to cope with ToA prediction.

In its general form, the block diagram of a CapsNet architecture nests a capsule representation in cascade to standard convolutional layers, without pooling in between them, to learn novel representations in a lossless way. More formally, it consists of the following two elements:*Primary capsule*: this layer performs convolution aggregation via the so-called *capsule unit*$u_{i}$ ($i\in \{1,\cdots ,N_{PC}\}$ being the capsule index), corresponding to multiple combinations of the feature maps retrieved at the end of the convolution process. In their working principles, primary capsules provide an alternative form of convolutional layers: the main difference is that, in this case, a vector-based output is computed rather than working with unitary depth. As such, convolution-based processing is performed by each capsule, which is driven by an appropriate set of kernels and relative stride.*Digit capsule*: at this point, the agreement among different capsules has to be estimated so that it is possible to preserve the spatial dependency between those block representations with highest relevance. This concept is mathematically encoded via the weight opinion matrix $W_{ij}$, with $j\in \{1,\cdots ,N_{class}\}$ being the number of classes, each with vector-based output of dimension $N_{DC}$. Hence, every capsule is judged by means of $N_{class}$ opinions $u_{j|i}$, also called local digit capsules, to be computed as
(2)$$u_{j|i}=W_{ij}u_{i}$$From these local representations, a further level of feature combination is added in a spatially dependent manner, by following the routing-by-agreement protocol [29]. This procedure, also called dynamic routing, introduces the concept of *agreement*, i.e., how much the individual digit capsules agree with the combined one. The level of agreement is numerically quantified by the weight routing matrix $R_{ij}$ via the coupling coefficient
(3)$$c_{ij}=\frac{e^{R_{ij}}}{\sum_{c=1}^{N_{class}}e^{R_{ic}}}$$As such, the final digit capsule $s_{j}$ is given by $s_{j}=\sum_{i}c_{ij}u_{j|i}$. As in traditional convolutional layers, activation is required to ensure that digit capsules with low opinions shrunk to zero, since they do not convey meaningful information. However, the vector-based output of the capsules requires ad hoc functions to fulfill this task: the squashing function
(4)$$v_{j}=\frac{\left|\right|s_{j}{\left|\right|}^{2}}{1+\left|\right|s_{j}{\left|\right|}^{2}}\frac{s_{j}}{\left|\right|s_{j}{\left|\right|}^{2}}$$
was purposely proposed in [28] to address it, where $s_{j}$ and $v_{j}$ are the input and output of the *j*-th convolutionally operated capsule. The quantity $u_{j|i}\;\cdot \;v_{j}$ finally yields the actual measure of agreement, i.e., the higher this product, the more preference is awarded to the corresponding primary capsule $u_{i}$. At this point, an iterative algorithm can be called to update the routing matrix, until the desired level of agreement is reached and the sought $N_{class}\times N_{DC}$ digit capsule block can be derived, which serves as the output layer for the entire neural network. Finally, it is sufficient to calculate the $\ell_{2}$ norm of each of the $N_{class}$ rows to obtain a corresponding value of the output probability associated to each single class.

For AE-related problems, just two classes can be considered, i.e., noise and AE signals: in this case, a high value of the output probability $p_{AE}$ for class “AE signal” indicates that the input instance is most likely to contain a true AE event, whereas low values can be seen as indicators of noisy input.

An overview of the proposed CapsNet architecture for AE signal processing is graphically summarized in the left-hand side of Figure 4. The initial convolutional block consists of two Conv1D layers without pooling, activated by ReLU and with dimensions (500,64,9,2) and (250,128,9,2), respectively. At the output of the convolutional layer, 128 feature maps of 125 samples each are computed: these feature maps are passed to the primary capsule layer. Here, 78 primary capsules of 8 feature maps each are created, and then processed via capsule operations via kernels of size 9 and stride equal to 3. Dynamic routing is then performed, yielding to $N_{class}$ different digit capsules with a vector size equal to 8. A last stage in which the $\ell_{2}$ norm is applied to each row of the digit capsule block returns the two desired class probabilities (noise and AE).

In terms of model complexity, the proposed CapsNet architecture requires 301,952 parameters and allocates a memory space of 375 kB, a quantity which is 4.2× lower and 2.25× bigger than the ones required by the original and small CNN, respectively. Once again, the Adam optimizer (with a learning rate of 0.001) was used for training the model for a total of 15 epochs.

#### ToA Retrieval from CapsNet: The CapsNetToA Architecture

Determining ToA with CapsNet is a two-step process. Indeed, in its definition, CapsNet acts a classifier for the input batch of data, meaning that it can only predict whether the current instance is most likely to contain low-entropy signal content (high probability) or rather noisy data (low probability). Therefore, a dedicated logic is implemented to extract one single time value out of the class probability distribution $p_{AE}$. Hereinafter, the entire processing flow, encompassing both CapsNet and the time retrieval logic, is named CapsNetToA.

To this end, an approach similar to the one suggested in [27] is adopted. The idea is to split the entire waveform of 5000 samples into smaller and overlapped windows, each of them identified by a unique time stamp taken as the central value of the corresponding time span. For every segment, a probability value is returned; the cumulative trend in the probability distribution can be easily obtained by concatenating, in time, the predictions related to subsequent windows.

The rationale is that the probability curve is expected to assume a low value until the signal statistics do not change. Then, when the first window containing the wave arrival is processed, the curve increases progressively, reaching its maximum (in the ideal case, unitary probability) for the exact window centered on the actual ToA.

On a first attempt, one may resort to statistical tools, such as pick-peaking or thresholding functions, to retrieve ToA as the first peak probability value. However, such a simple approach might suffer from several drawbacks, which can be listed as follows: it presents poor generalization capabilities, in the sense that the selection of a threshold or benchmark value is strictly application and environment dependent; as such, it badly conjugates with the critical variability of AE scenarios. The second reason relates to the impossibility of accurately retrieving very early onset times, i.e., the ones below or almost equal to the window length, for which the true peak probability value is unavoidably missed. In this case, in fact, all the initial windows will output a nearly unitary value and, thus, a criterion based on the leading peak selection unavoidably will estimate ToA from secondary signal arrivals.

Conversely, ML solutions can inherently handle all these sources of complexities in a very efficient and user-transparent way. For this reason, a second NN block is stacked in cascade to CapsNet to retrieve ToA from the output probability history yielded at the end of the capsule processing. It is worth observing that, in ToA terms, the problem is exactly analogous to the one faced to estimate ToA via CNN while working directly with time series data. The main difference is that, for this second scenario driven by CapsNet, probability functions are available as inputs. Coherently, in the approach presented in this work and which is novel with respect to the one in [27], it is suggested to employ the same small CNN as “ToA logical retrieval” block (see Figure 4) in a completely agnostic and general purpose manner.

The parameters of CapsNetToA were configured as follows. Assuming an operative sampling frequency of 2 MHz, the selection of a window size of 500 samples with stride equal to 10 imposed a lower bound of 5 μs to the ToA resolution. This value is compliant with the time resolution admitted for the prospective applications, where ToA usually settles around hundreds of microseconds.

## 3. Experimental Validation: A Numerical Framework

The effectiveness of the designed models was tested within the framework of a metallic aluminum plate, which is frequently exploited as a benchmark scenario for ToA estimation. Firstly, a preliminary phase of dataset generation was performed to train the models, whose accuracy in prediction was then assessed by comparison with ground truth labels. Noteworthy, this initial validation is of critical importance to validate the robustness of the AI solutions with respect to reference statistical methods, especially to observe how performances can scale in the presence of noise levels.

### 3.1. Dataset Generation

As is widely recognized, DL models require a large amount of data to be trained on to avoid erroneous predictions. Moreover, since we are dealing with prediction problems, the same data also need to be labeled. However, labeling a massive amount of experimental data is, unfortunately, practically unfeasible. Alternatively, analytical simulations could be adopted to rapidly generate the labeled dataset. In this case, as anticipated in Section 1.1, it is possible to exploit the fact that acoustic emissions travel along the mechanical medium in the form of GWs, for which the propagation pattern through the mechanical medium is well known, and a numerical simulator can be implemented.

In our scenario, a square aluminum plate with nominal thickness of 3 mm and length of 1000 mm was taken as reference, while a Gaussian modulated pulse with central frequency of 250 kHz was assumed to simulate the effect of acoustic sources: in this frequency range, only the A0 and S0 modes characterize the propagation behavior, a condition which is desired to minimize the detrimental effect of multi-modality. Multi-path effects due to reflections and reverberations were considered as well with a ray-tracing approach, purposely written in MATLAB©. Each time series consisted of 5000 samples acquired at a theoretical sampling frequency of 2 MHz: these quantities were chosen to be compatible with commercial off-the-shelf sensors for AE monitoring.

More in detail, the signal generation procedure followed the subsequent steps:**Traveling distance selection**: theoretically, the number of possible propagation distances to be explored between the AE location and the receiving point is infinite. However, by exploiting the symmetry of the structure ensured by its isotropic nature, the number of useful configurations can be reduced by a large extent. A square area of 5 × 5 positions circumscribed to the top east corner of the plate was allocated to AE receivers, while a total amount of 10 × 5 AE actuation points were uniformly distributed in the left half of the plate.**Noise level variation**: since the primary objective of the proposed NN alternatives is to surpass the poor estimation capabilities of reference statistical methods in the presence of noise, Gaussian noise of increasing magnitude was progressively added to the acoustic wave by sweeping the SNR from 30 dB down to 1 dB, in steps of almost 1 dB. Despite the fact that the nature of the background noise of real AE signals can indeed differ [30], additive white stationary noise (such as the one generated by electronic components) can be assumed to be the main source of SNR degradation and, consequently, was used to simulate noisy AE scenarios in this study.**Pre-trigger window variation**: in real AE equipment, the starting time for data logging is triggered by the incoming wave, e.g., once it exceeds a predefined energetic threshold. However, being capable of acquiring also the moments leading up to the acoustic event is of vital importance for appropriate AE signal characterization. As such, sensors are programmed to preserve memory of the pre-trigger signal history, known as the pre-trigger window. This quantity might change widely, from hundreds to thousands of samples, depending on both the application scenario and the employed electronics.Although representing a deterministic parameter that does not strictly depend on the physical phenomenon at the basis of acoustic wave propagation, the pre-trigger time actually plays a crucial role during the learning stage. This observation means that, theoretically, a one-to-one correspondence should exist between one model and one pre-trigger window. This aspect not only requires time and extra computing effort, due to the fact that a new training phase must be entailed whenever a change in the network configuration occurs, but it is also not viable in practical scenarios. Therefore, a data augmentation procedure was encompassed to favor the generalization capability of the neural network models.To this end, acoustic signals were initially generated with a fixed pre–trigger window of 500 samples, that represents a reasonable choice for typical scenarios. Then, one time-lagged version of each signal was derived by adding randomly from 500 to 2000 samples to the initial pre-trigger window. Since the total number of samples in the time history is limited to 5000, these forward shifts required additional $N_{pre}$ samples to be concatenated with the initial portion of the signal, while disregarding the final $N-N_{pre}$: to avoid both discontinuities and alterations in the statistical properties, the extra portion of the signal to be added was generated in form of a white noise term drawn from a Gaussian distribution, whose variance was taken to be coincident with the one estimated for the first 400 samples in the original pre-trigger window.Another batch of data was also generated, comprising signals with an increased pre-trigger time beyond 2500 samples, and was entirely used during the testing phase in order to probe how the neural networks could behave with respect to unforeseen delays in the signal.**Label generation**: when Lamb waves are to be characterized, it is difficult to give an unambiguous definition of their time of arrival due to dispersion and multi-modality. For this reason, rather than adopting a labeling approach based on the propagation theory, a different strategy was undertaken in this work. In particular, we exploited the fact that AIC inherently provides very accurate ToA estimations when the SNR is high. As such, the label attached to each time series was taken from the output yielded by AIC when applied to noise-free signals.

A total amount of $N_{inst}$ = 60,000 signals was generated via exhaustive combination of all the possible configurations between the propagation distance, noise level and pre-trigger window: 80% of them were used for training, 10% for validation, and the remaining 10% for testing. Each time series was then normalized and mean removed. Some exemplary signals collected at the end of the dataset generation phase are plotted in Figure 5.

### 3.2. Performance Metrics

Since “true” labels are available, the simplest methodology to assess the accuracy of the models is to quantify the error between predicted and actual ToAs. This strategy is efficient since it allows also to probe the accuracy of AIC in noisy scenarios: indeed, once applied to noise-corrupted variants of the same data, AIC predictions might deteriorate proportionally to the level of added perturbations.

The canonical expression of the root mean square error (RMSE),
(5)$$\mathrm{RMSE}=\sqrt{\frac{\sum_{i=1}^{N_{test}}{\left(ToA_{i}-{T\hat{o}A}_{i}\right)}^{2}}{N_{test}}}$$
was resorted to as an accuracy indicator: $ToA_{i}$ is the true value, ${T\hat{o}A}_{i}$ is the current prediction, and $N_{test}$ is the number of instances used during testing.

### 3.3. Results

The ToA estimation results for test data with a pre-trigger window below 2500 (i.e., the one used for training) are displayed in the left panel of Figure 6: they are given in terms of RMSE (log scale) for increasing SNR. From this figure, two different trends in the reported errors are evident. Firstly, the error profile of AIC is inversely proportional to the hidden noise level affecting the signal; moreover, it is characterized by a slowly decaying trend with a smoother profile. The performance results of the four approaches are in the same order of magnitude only for SNR equal to 30 dB, i.e., when, as discussed before, the signal statistics are well defined and easy to be identified, even via conventional processing tools.

Secondly, in the comparison between the AI approaches, the largest CNN shows an almost stable error around 4.5 μs, irrespective from the specific SNR apart from a negligible increment for very unfavorable noise levels. It is worth noting that the distillation operation (small CNN) is less performative for SNR below 5 dB but very effective for all the remaining SNRs. Finally, the curve error of CapsNetToA is similar to the one discussed for small CNN with slightly higher errors at high SNRs.

A graphical depiction of the output collected from the different approaches is displayed in Figure 7, which shows a zoom in the 0.1–0.4 ms window with drawn star markers indicating the predictions obtained from AIC (blue), CNN (orange), CapsNetToA (red) and small CNN (green), superimposed to the true label (yellow diamond). As can be observed, the estimates are considerably accurate in the left panel (SNR = 30 dB) for all the methods, while AIC completely fails in the case of important noise levels (same signal, but SNR = 10 dB) depicted in the right-hand side.

The performance achieved by varying the pre-trigger windows is reported in the right-hand side of Figure 6. As can be seen, no change is observable for AIC, owing to its time invariance. Conversely, an abrupt loss of accuracy affects the biggest CNN model, whose RMSE error maintains the same profile at the expense of a 20× increase in the magnitude. Such behavior can be attributed to over-parametrization problems as well as to poor generalization capabilities of the CNN due to the huge amount of parameters to be learned with respect to the actual amount of instances used in the training phase and the complexity of the problem at hand. In further detail, this is due to the fact that the number of learnable parameters largely exceeds the total amount of training instances. A second reason for such difference in CNN performances consists in potential *overfitting*, i.e., over-adaptation to the training dataset leading to a lack of capability to generalize during testing. Consequently, the biggest CNN model may not adequately capture the hidden ToA information in AE signals.

Conversely, as shown in Figure 6, the smallest CNN model is not prone to overfitting or over-parametrization, thanks to its more appropriate balancing between the number of trainable parameters and available data. Therefore, the distilled model is capable of generalizing from the trend hidden within training data, which is actually the expected goal of the neural network, rather than memorizing the training data themselves as happens with the largest CNN. Proof is in the fact that the small CNN model is still capable of providing coherent results, once again showing the best results among all the considered methods. For these reasons, only the small CNN model and CapsNetToA are taken into consideration in the following analyses, due to their better performance in terms of memory footprint, accuracy and generalization with respect to the length of the pre-trigger window.

Finally, the computational complexity of the designed NN models was also evaluated in terms on execution time in order to assess their actual portability on edge devices. To this end, when tested on a 1.8 GHz dual core Intel^®^ core i-5 processor, the average inference time for a single signal was measured to be equal to 8.63 ms, 4.79 ms and 39.61 ms for CNN, small CNN and CapsNetToA (comprehensive, in this case, of both CapsNet and the time retrieval logic), respectively. As can be observed, the computational time nearly halves while moving from the biggest to the smallest CNN, while CapsNetToA shows the longest execution time. This is due to the fact that this architecture requires the ToA output probability curve to be reconstructed from several overlapped windows of the input signal, an operation which imposes CapsNet to be executed multiple times. For the specific CapsNet design configuration considered in this work, 1200 sliding windows need to be processed, each of them asking for nearly 30 μs. Note that such an amount of time can be reduced either by reducing the number of windows or by changing the length of the window itself.
In line with these results, it is worth saying that, even if the performance might scale when the same algorithms are deployed on embedded devices depending on the frequency clock of the featured processor, the above reported execution times are compatible with the near-sensor implementation of the investigated models.

## 4. Experimental Validation: ToA for Acoustic Source Localization

In this section, experimental data for a laboratory aluminum plate, companion to the simulated one used for the sake of preliminary validation, are exploited to assess the suitability of the trained models to cope with acoustic source localization problems driven by in-field data.

### 4.1. Materials: The AE Equipment

The acoustic waves are actuated in form of guided elastic waves via a tone burst of two cycles, central frequency of 250 kHz and nominal amplitude of 0.6 Vpp, by means of an arbitrary waveform generator Agilent 33220A: its output was passed through a gated power amplifier RITEC GA-2500A (100× signal magnification) and then connected to an actuator, a Murata piezoelectric ceramic disk. Signals were acquired by a miniaturized acquisition system, the so-called piezoelectric ‘sensor node (SN)’ able to acquire, pre-process and characterize AE signals for real-time continuous monitoring. The core of the SN is the STMicroelecronics STM32F3, a low voltage 32-bit mixed signal microcontroller (MCU), which integrates both digital signal processing and floating-point unit instructions. The MCU features 40 kB SRAM and 256 kB FLASH memory, enough for the storage of data acquired by three AE sensors. The MCU embeds high precision analog components, such as four analog-to-digital converters (ADCs) with a resolution up to 12 bit via successive approximation and embedded voltage reference, four rail-to-rail input/output, low offset voltage programmable gain amplifiers (PGAs) and two 12 bit digital-to-analog converter (DAC) channels, which allow to obtain the maximum output swing. The frequency response of the device provides stable behavior from 10 kHz and 600 kHz, with a maximum attenuation of 3 dB. A complete description of the electronics is detailed in [31,32]. Each device of the network features up to three piezoelectric acquisition channels S1, S2, and S3, which passively acquire signals. Three piezoelectric sensors G150 whose frequency range lies between 60 kHz and 400 kHz were deployed at three corners of the plate, as schematically drawn in Figure 8, and a picture of the setup is presented in Figure 9.

### 4.2. Methods

#### 4.2.1. Objectives and Testing Procedures

Two main objectives were pursued with these experiments: (i) assess the time picking performance against operative disturbances responsible for important noise levels, and (ii) evaluate the suitability of the devised neural network architectures to deal with source localization. Nine different points (star markers in Figure 8) were selected for actuation, which are uniformly spaced in a square area with length equal to 0.3 m. For each point, experiments were repeated three times with a constant sample window of 5000 samples, pre-trigger window of 1500 samples and sample rate equal to 2 MHz. To evaluate the effect of noise, and following the testing procedure described in Section 3.1, white Gaussian noise was added to the gathered time series by sweeping the SNR in the interval of 2–20 dB at integer steps of 4 dB.

#### 4.2.2. Localization Algorithm

To achieve successful object localization in 2D environments, a minimum network density of three sensing elements with known position has to be deployed on the monitored structure. Among the various strategies, the triangulation method proposed in [33] was leveraged in this work for its geometrical simplicity and well-proven functioning for isotropic/homogeneous structures, such as the one offered by the considered test-bed.

A graphical rendering of the triangulation problem is pinpointed in Figure 10, where a generic plate is instrumented with a network of three sensors, namely S1, S2 and S3. If S2 serves as a reference sensor for the network, $\theta_{1}$ and $\theta_{3}$ represent the relative orientation of sensors S1 and S3 with respect to a horizontal axis crossing S2 and aligned along the longitudinal dimension of the plate; similarly, $D_{12}$ and $D_{32}$ indicate the spatial distance between the two pairs of sensors, S1–S2 and S3–S2.

Now, if an acoustic event (red thunderbolt with label AE) occurs at a generic point of the structure, the triangulation algorithm aims at retrieving the set of polar coordinates (*d*, θ) uniquely identifying the acoustic source in the space. $d=d_{2}$ corresponds to the sought S2–AE distance, while $\theta_{2}=\theta$ indicates where, among all the possible directions, the acoustic signal comes from. Once generated, the acoustic wave propagates over the structure and strikes the three sensors in three different instants of time due to the different AE-to-sensor distance: by application of the ToA estimation strategies investigated before, a guess of these onset times can be formulated. Let us denote them as $ToA_{1}$, $ToA_{2}$ and $ToA_{3}$ for S1, S2 and S3, respectively. Unfortunately, these quantities do not coincide with the physical time $t_{i}=d_{i}/c_{g}$ taken by the wave to travel the corresponding distances $d_{i}$ at a wave speed $c_{g}=5205$ m/s (related to the fastest propagating mode, the S0 one, at a frequency of 250 kHz); they rather represent a cumulative sum between $t_{i}$ and the event actuation time $T_{0}$, which is common to all the sensors but not known to the system.

Thus, even if geometrically well-posed, the estimation of *d* from the mere knowledge of $ToA_{2}$ is not practicable. However, by computing the DToA $\delta_{i2}$ out of the absolute ToA between the reference sensor S2 and the two remaining nodes, the $T_{0}$–independent quantities
$$\begin{array}{c} \delta t_{32}=ToA_{3}-ToA_{2}=\left(t_{3}+T_{0}\right)-\left(t_{2}+T_{0}\right)=t_{3}-t_{2}\\ \delta t_{12}=ToA_{1}-ToA_{2}=\left(t_{1}+T_{0}\right)-\left(t_{2}+T_{0}\right)=t_{1}-t_{2} \end{array}$$
can be obtained for sensor pairs S2–S3 and S2–S1, respectively, which coincide with the physical differences between the true wave propagation times. Analogously, the spatial difference between the traveled distances can be formulated as
$$\begin{array}{c} \delta d_{32}=d_{3}-d=c_{g}\delta t_{32}\\ \delta d_{12}=d_{1}-d=c_{g}\delta t_{12} \end{array}$$

The mathematical problem can thus be solved by means of the linear system:(6)$$\left\{\begin{array}{c} d_{3}=d+c_{g}\delta t_{32}\\ d_{1}=d+c_{g}\delta t_{12} \end{array}\right.$$
which, in this form, is not solvable since three unknowns but only two equations are available. To overcome this issue, it is sufficient to apply simple trigonometric relationships to the geometrical scheme in Figure 10: in particular, the Carnot’s theorem states that
(7a)$$\begin{array}{c} d_{3}^{2}=d^{2}+D_{32}^{2}-2dd_{32}\cos\left(\theta_{3}-\theta \right) \end{array}$$
(7b)$$\begin{array}{c} d_{1}^{2}=d^{2}+D_{12}^{2}-2dd_{12}\cos\left(\theta -\theta_{1}\right) \end{array}$$

Hence, by taking the square power of both sides in Equation (Equation 6) and plugging Equation (7a,b) in its first and second rows, respectively, the system can be rewritten only in terms of *d* and θ, which are the sought outputs of the localization process:(8)$$\left\{\begin{array}{c} d=\frac{1}{2}\frac{D_{32}^{2}-c^{2}\delta t_{32}^{2}}{c_{g}\delta t_{32}+D_{32}\cos\left(\theta_{3}-\theta \right)}\\ d=\frac{1}{2}\frac{D_{12}^{2}-c^{2}\delta t_{12}^{2}}{c_{g}\delta t_{12}+D_{12}\cos\left(\theta -\theta_{1}\right)} \end{array}\right.$$

Algebraic manipulation of the system in Equation (Equation 8) yields the auxiliary solving equation for θ to coincide with:(9)$$r\sin(\theta +\theta_{a})=K$$
with
$$\begin{array}{c} G=\frac{D_{32}^{2}-c_{g}^{2}\delta t_{32}^{2}}{D_{12}^{2}-c_{g}^{2}\delta t_{12}^{2}}\\ g_{1}=GD_{12}\cos\theta_{1}-D_{32}\cos\theta_{3}\\ g_{2}=GD_{12}\sin\theta_{1}-D_{32}\sin\theta_{3}\\ K=-c_{g}\left(G\delta t_{12}-\delta t_{32}\right)\\ r=\sqrt{g_{1}^{2}+g_{2}^{2}}\\ \theta_{a}=\tan^{-1}\left(\frac{g_{1}}{g_{2}}\right) \end{array}$$

Finally, *d* can be computed back from the system in Equation (Equation 8). By knowing θ and *d*, the estimated AE source position $\hat{P}\left({\hat{x}}_{P};{\hat{y}}_{p}\right)$ can be easily derived as
(10)$$\left\{\begin{array}{c} {\hat{x}}_{P}=x_{S_{2}}+d\;\cdot \;\cos\theta \\ {\hat{y}}_{P}=y_{S_{2}}+d\;\cdot \;\sin\theta \end{array}\right.$$
where $x_{S_{2}}$ and $y_{S_{2}}$ represents the 2D coordinates of the reference sensor S2.

#### 4.2.3. Performance Evaluation Procedure

All the quantities appearing in Equation (Equation 9), apart from $\delta t_{12}$ and $\delta t_{32}$, are deterministic once the sensor network configuration is defined. As such, achieving good localization capability is, inversely, a synonym of accurate onset time estimation and offers, in these terms, a powerful means to assess the quality of the time picking activity of the devised NN models. Note that, in this operative setting where no synchronization is present between the actuation and the reception components of the monitoring network, no possibility exists to define a true ground truth due to the uncertainties implied by the experimental setting.

During testing, three ToA estimators were considered for the purpose: besides the small CNN model and CapsNetToA trained on the synthetic data, AIC was also applied from the pool of conventional statistical approaches. The Euclidean distance $d_{p}=\left|P-\hat{P}\right|$ between the true $P(x_{P};y_{p})$ and estimated $\hat{P}\left({\hat{x}}_{P};{\hat{y}}_{p}\right)$ AE position returned by Equation (Equation 10) were computed to assess the accuracy for source localization task. Moreover, the collected signals were corrupted with white Gaussian noise of increasing entity, i.e., by gradually reducing the SNR from 20 dB to 2 dB. In this way, it was possible to investigate how the same model could perform in remarkably harsher propagating environments, as they could appear in real AE scenarios subjected to varying but not predictable noisy conditions.

### 4.3. Results

Outcomes from localization experiments are drawn in Figure 11: for each excitation point, three markers are included, corresponding to as many tests in this configuration. Additionally, it is worth specifying that points indicating a constant error of $d_{P}=1$ m are included to identify all those cases in which the estimated coordinates are not compatible with the physical solutions for the structure at hand (i.e., negative or larger than 1.41 m distances, which is the maximum propagation length for this plate). Figure 12 provides an example of time domain signals for excitation point 5 in two different noisy conditions: 20 dB on the left column and 8 dB on the right, with magnified ToA values in the second row.

First of all, one remark is worthy of attention, which is related to the difference in the experimental signals with respect to the synthetic ones employed for training (see comparison between Figure 5 and Figure 11). In fact, in the real setting, the effects of the sensor transfer function as well as the detrimental ones due to attenuation, multiple reflections and propagation in the physical medium might lead the envelope of the acquired signals to vary in a significant way. As such, being capable of obtaining an accurate prediction on real signals starting from a simulated dataset can be seen as a more severe test to be passed.

Going deeper into the results, the NN methods provide more consistent estimates, irrespective of the single excitation point and just showing a minor increment starting from SNR = 4 dB, where the errors increase up to 30 cm for 2 dB, which represents a challenging working condition for AE equipment. Conversely, $d_{P}$ reduces to a large extent in all the remaining noisy configurations, the CapsNetToA model being the most accurate estimator with an average error of 5 cm, followed by small CNN, whose average error is less than 8 cm.

Moreover, it is possible to observe that the quite similar localization pattern shown by small CNN, which tends to worsen the lower the SNR, is a consequence of the noise generation procedure exposed in Section 3.1, according to which the same signal, but with different additive noise, is processed. For the same reason, even if less pronounced, an analogous trend characterizes also AIC and CapsNetToA.

The slightly better performance of CapsNetToA with respect to the small CNN can be attributed to the superior generalization capability of the first solution, which better handles the discrepancy between the actuated pulse and the synthetic one used in the training. This generalization capability is a very desirable property in practical scenarios, where the actuated pulse is unknown.

Moreover, the same plots show that, despite AIC being highly performative in low noise conditions, as proven by a maximum deviation of 7 cm for the 20 dB configuration, the introduction of high noise levels leads AIC to completely fail ($d_{P}=1$ m) in multiple positions (e.g., $P_{1}$, $P_{2}$, and $P_{3}$).

The above considerations confirm the trends already reported in Figure 6, showing that AIC remains a robust and competitive strategy for AE signal characterization for relatively low noise values affecting data, whose drawbacks in dealing with poor SNRs can be overcome via AI approaches, and which can still achieve a satisfactory level of precision in the identification of the acoustic source.

## 5. Conclusions

In this work, innovative methods taken from the AI processing field are presented for ToA estimation in AE-generated signals. The proposed solutions, based on a multi-layer CNN and on a capsule architecture with dedicated time retrieval logic, have the peculiar advantage of providing consistent results, even in the presence of significantly low SNRs. When tested on both a synthetic dataset generated for the characterization of a square aluminum plate and for source localization in laboratory conditions, they attained 10× more accurate results than the AIC algorithm, which can be considered a standard in the field. In particular, among the proposed NN methods, the novel CapsNetToA scored the highest results thanks to its correlation-based approach, achieved via the unique concept of the capsule unit, which allows spatial dependencies between data to be preserved. Secondly, by replacing pooling layers, which are the primary source for loss of information in conventional CNN architectures, with peculiar vector-based output layers, it can implicitly generalize better to out-of-distribution data and gain higher immunity to noisy outliers. Future work will be devoted to embedding of the same algorithms on edge devices, for the sensor-near retrieval of the ToA as required in practical scenarios. Furthermore, additional models and related experimental results will be investigated, in which the frequency content of the excited signals will be changed according to the different spectral bands used in AE testing.

## Figures and Tables

**Figure 1 sensors-22-01091-f001:**
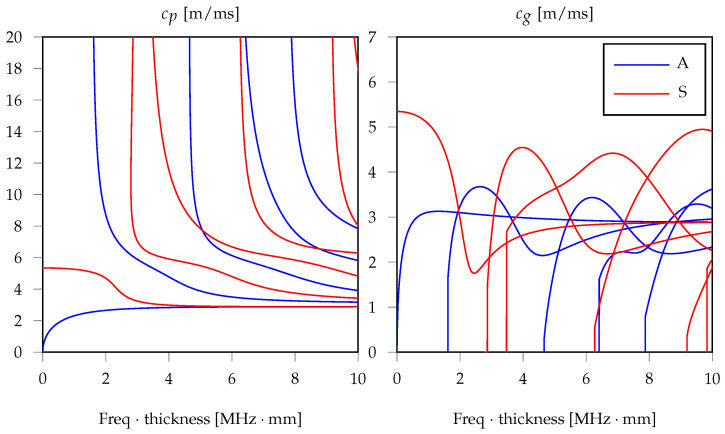
Dispersion curves for an aluminum square plate with 1mm thickness and infinite extension: phase velocity on the **left** and group velocity on the **right**.

**Figure 2 sensors-22-01091-f002:**
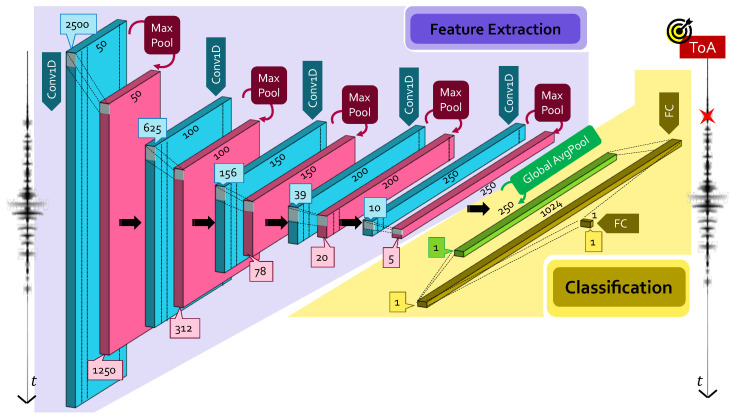
Proposed CNN for ToA estimation in 5000-long time series. The quantities reported in the blue and pink boxes indicate the dimensions of the output features of the corresponding layer.

**Figure 3 sensors-22-01091-f003:**
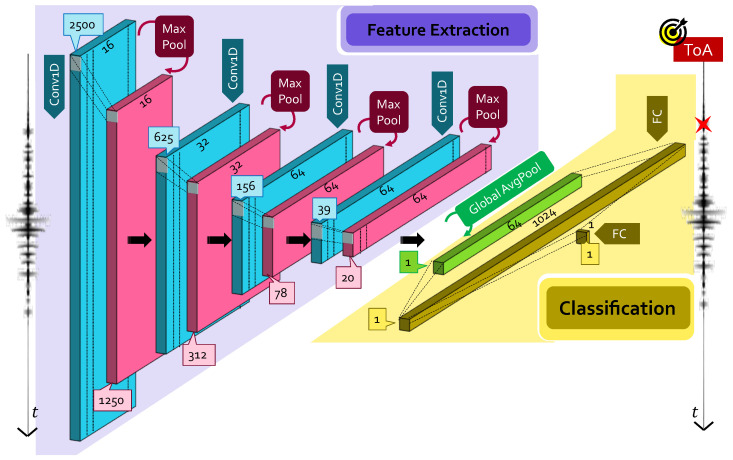
Proposed small CNN for ToA estimation. The quantities reported in the blue and pink boxes indicate the dimensions of the output features of the corresponding layer.

**Figure 4 sensors-22-01091-f004:**
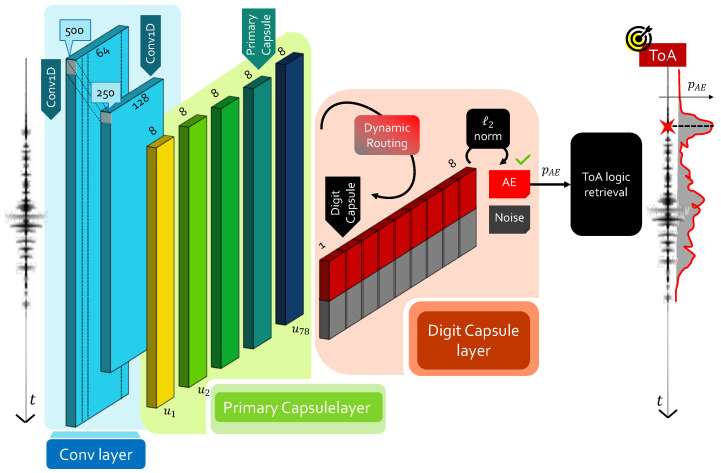
Proposed CapsNetToA architecture for ToA estimation.

**Figure 5 sensors-22-01091-f005:**
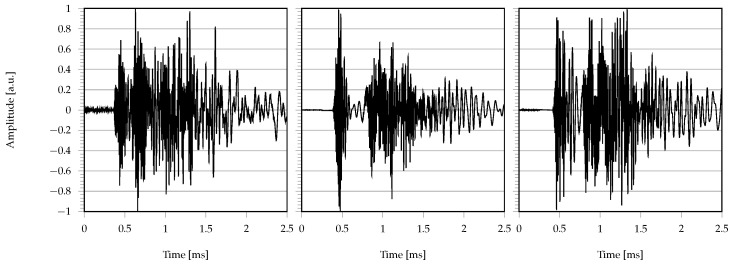
Example of synthetic signals generated with the ray–tracing algorithm.

**Figure 6 sensors-22-01091-f006:**
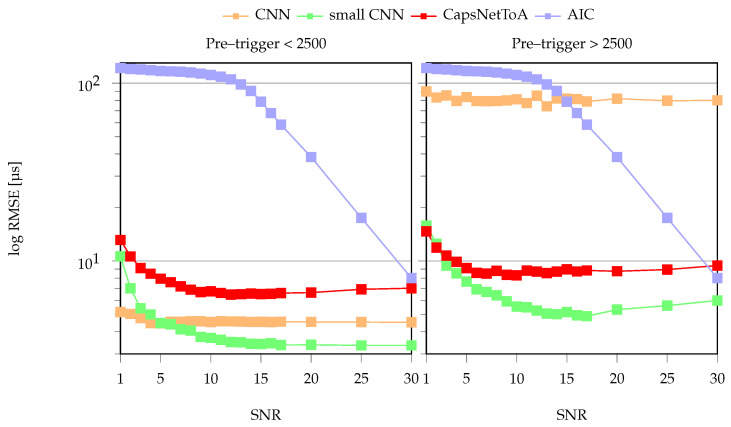
RMSE error for ToA estimation on synthetic test dataset for varying SNRs: Pre–trigger window used for training (i.e., window lower than 2500 samples) on the left and increased pre–trigger window on the right.

**Figure 7 sensors-22-01091-f007:**
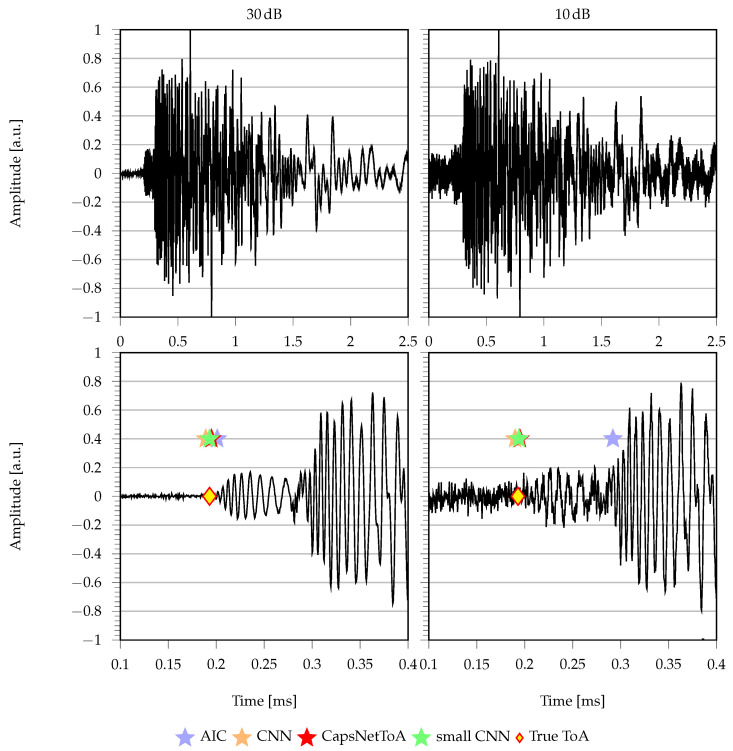
ToA predictions for AIC (blue), CNN (orange), CapsNetToA (red) and small CNN (green) with synthetic dataset: (**left column**) SNR = 30 dB and (**right column**) SNR = 10 dB, with second row depicting a magnified region of the time interval where ToA is located.

**Figure 8 sensors-22-01091-f008:**
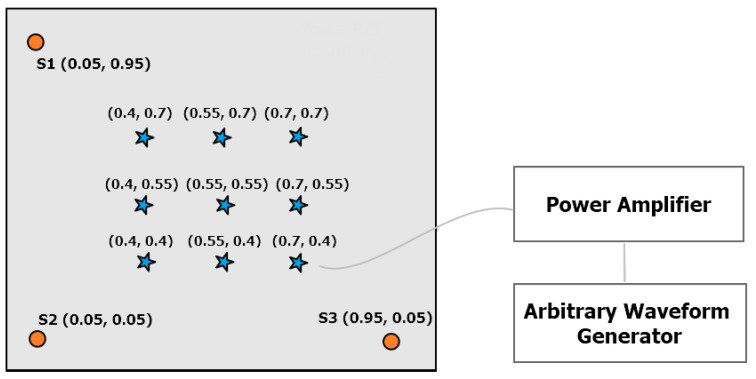
Sensor deployment diagram on the metallic laboratory plate used for the sake of source localization.

**Figure 9 sensors-22-01091-f009:**
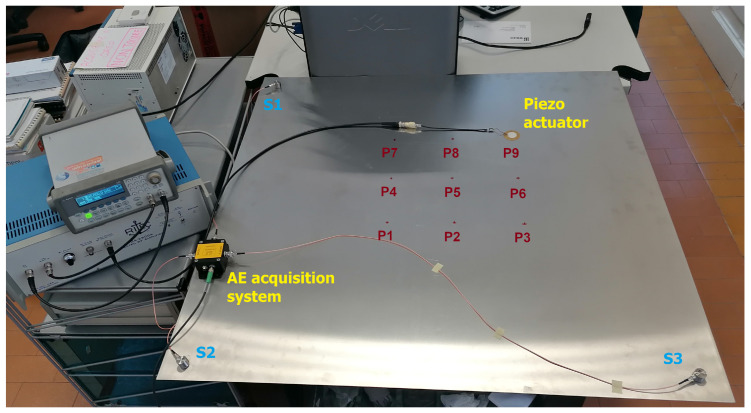
A photo of the experimental setup.

**Figure 10 sensors-22-01091-f010:**
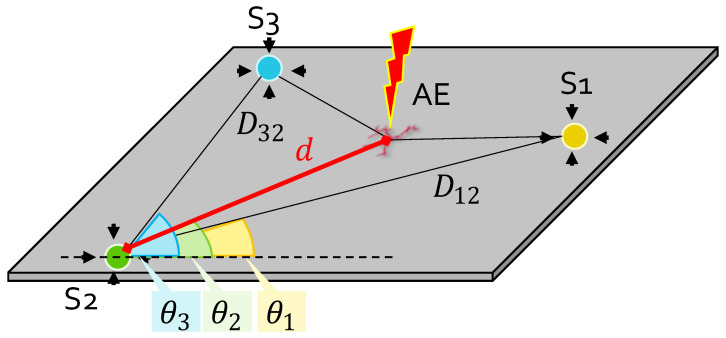
Geometrical representation of the adopted localization model via a three sensor array: the objective is to estimate the AE-to-sensor distance *d* and its relative direction θ.

**Figure 11 sensors-22-01091-f011:**
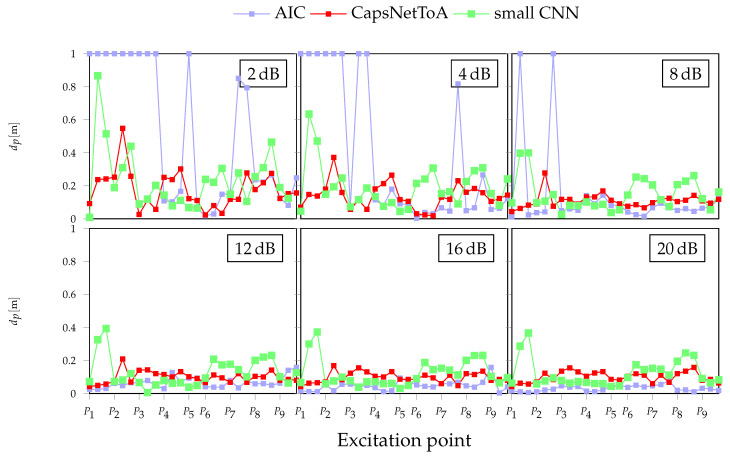
Absolute errors for acoustic signal localization on a laboratory metallic plate under various noise levels.

**Figure 12 sensors-22-01091-f012:**
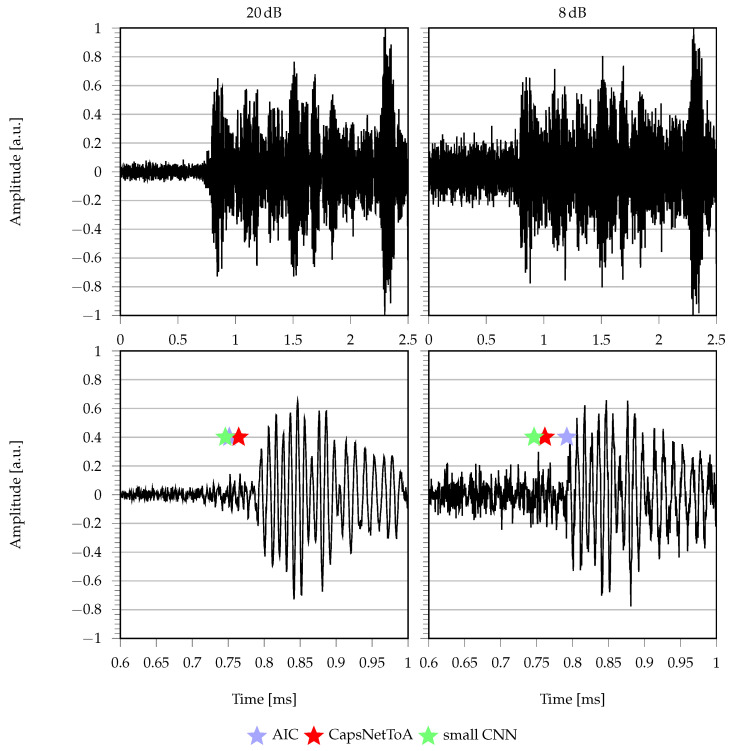
ToA predictions for AIC (blue), CapsNetToA (red) and small CNN (green) for excitation point 5: (**left column**) SNR = 20 dB and (**right column**) SNR = 8 dB, with second row depicting a magnified region of the time interval where ToA is located.

## Data Availability

Not applicable.

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
