# Peer review of "Deep Learning Approaches for Robust Time of Arrival Estimation in Acoustic Emission Monitoring"

_sensors, 2022, doi:10.3390/s22031091_

Round 1

Reviewer 1 Report

The reviewed manuscript presents an estimation method of the time of arrival (ToA) in Acoustic Emission (AE) signals. The work is interesting. The manuscript is well organized. However, some issues should be checked and improved in new possible re-submission or revised version of the manuscript.

  1. It seems that training dataset is generated by the simulation model. The noise in the actual detection process may be more complex than that of the simulated noise. So the reliability of the training dataset is in doubt. Furthermore, author himself agrees this view in this paper (Line 567-574).
  2. In the Conclusion Section, the reason of the slightly better performances of CapsNetToA with respect to the Small CNN is too shallow. Author should dig deeper the advantages of the CapsNetToA to explain the reason.
  3. In Line 445, the Figure Number may be wrong. Author should check the whole manuscript to modify similar mistakes.

Reviewer 2 Report

The authors provide a quite interesting approach to apply ML models for AE monitoring.
Some technical details about the receiver characteristics are missing, led to a difficulty to understand if the solution can be applied within edge nodes, as claimed many times in the paper.
In that regard, the authors does not provide details or a comparison on the computational time of the different models and their suitability for edge computing.
The state-of-the-art could be enlarged to take into account other techniques that lead to an increase of accuracy, to highest robustness with regards to SNR levels and to (probably) shorter computational times.

Introduction

Example in literature shows that growth of cracking phenomena can be detected using AE also by exploiting wavelet-based and convolutional-neural-network-based algorithms.

ToA is used also for indoor positioning system and is proven to provide higher results with proper designed linear chirp signals, that increase the range of the acceptable SNR levels of the signal.  
The chirp cross correlation shows a very sharp and easy to recognize peak (https://doi.org/10.1109/TIM.2018.2866358).

2.1.2

What are the drawbacks of the applied network model reduction ?

3.3
It seems counterintuitive that the smallest CNN, distilled from the biggest one, provides better results as shown in Fig. 6.
Please provide a detailed explanation of the behavior in addition to that provided in lines 453-458. It is not clear because the overfitting should apply only to the biggest CNN model and why well-known techniques for reducing its impact haven't been applied.

4.1
Analog to digital characteristics of the Sensor Node should be included in the paper, for the sake of readability, instead of refering another paper.

4.3
Small CNN, for every SNR value, provides a quite similar pattern, probably due to unbalanced training. Please provide some explanation or reply the test with a fine balance of the dataset.

Round 2

Reviewer 2 Report

Authors have addressed all my previous concerns. The manuscript is now worth publishing.